# Empiric Unsupervised Drifts Correction Method of Electrochemical Sensors for in Field Nitrogen Dioxide Monitoring

**DOI:** 10.3390/s21113581

**Published:** 2021-05-21

**Authors:** Rachid Laref, Etienne Losson, Alexandre Sava, Maryam Siadat

**Affiliations:** Laboratoire de Conception, Optimisation et Modélisation des Systèmes, LCOMS EA 7306, Université de Lorraine, 57000 Metz, France; etienne.losson@univ-lorraine.fr (E.L.); alexandre.sava@univ-lorraine.fr (A.S.); maryam.siadat@univ-lorraine.fr (M.S.)

**Keywords:** electrochemical sensors, calibration, long term drift, multiple linear regression, particles swarm optimization, in field nitrogen dioxide monitoring

## Abstract

This paper investigates the long term drift phenomenon affecting electrochemical sensors used in real environmental conditions to monitor the nitrogen dioxide concentration [NO_2_]. Electrochemical sensors are low-cost gas sensors able to detect pollutant gas at part per billion level and may be employed to enhance the air quality monitoring networks. However, they suffer from many forms of drift caused by climatic parameter variations, interfering gases and aging. Therefore, they require frequent, expensive and time-consuming calibrations, which constitute the main obstacle to the exploitation of these kinds of sensors. This paper proposes an empirical, linear and unsupervised drift correction model, allowing to extend the time between two successive full calibrations. First, a calibration model is established based on multiple linear regression. The influence of the air temperature and humidity is considered. Then, a correction model is proposed to solve the drift related to age issue. The slope and the intercept of the correction model compensate the change over time of the sensors’ sensitivity and baseline, respectively. The parameters of the correction model are identified using particle swarm optimization (PSO). Data considered in this work are continuously collected onsite close to a highway crossing Metz City (France) during a period of 6 months (July to December 2018) covering almost all the climatic conditions in this region. Experimental results show that the suggested correction model allows maintaining an adequate [NO_2_] estimation accuracy for at least 3 consecutive months without needing any labeled data for the recalibration.

## 1. Introduction

Life quality and human health are affected by air pollution especially in urban areas, where most of the population lives [1,2]. Nitrogen dioxide (NO_2_) is one of the major pollutant gases, and its emanation is mainly caused by traffic. Public environmental protection organizations ensure the NO_2_ quantification using sophisticated and high precise analyzers. These instruments are expensive and lodged in bulky stations which explain the poor spatiotemporal resolution of the air pollution monitoring networks [3,4]. Several research works revealed the low cost sensors potential for air quality monitoring systems [5,6,7]. Therefore, commercial systems have emerged to solve the above problem [8,9]. In fact, many regulatory organizations such as European commission and U.S Environmental Protection Agency recognized and encouraged the development of low cost technologies of air quality sensors in order to implement mixed networks including reference monitors and low cost sensors [10,11].

Electrochemical cell based sensor is a promising technology for air pollution monitoring [11]. This type of sensor provides high selectivity, low limit of detection, low power consumption and linear response to the target gas [12,13]. Most of low cost commercials air monitoring systems are based on this technology [14]. However, several issues impede these systems to provide data with quality similar to those obtained by the analyzers [15]. The mechanism of the sensor drifts is clearly presented in [16]. The main reason is that electrochemical sensors are affected by environmental factors, mainly the temperature and humidity [17,18]. In fact, temperature and humidity variation influence directly the sensors signal. On one hand, the chemical reaction depends on the temperature variation. On the other hand, high or low humidity rate can change the electrolyte volume inside the chemical sensors: high humidity rate may cause leakage of electrolyte, while low humidity dries the sensor electrolyte. Other obstacles are related to interfering gases and over time drifts caused by sensor aging. Several studies have been undertaken to understand and to deal with these issues. Mead et al. [19] considered the effect of the environmental factors and the interfering gases on electrochemical sensors for NO_2_, CO and NO concentration monitoring. They also addressed the short and long terms drifts. The authors assume that the change in baseline is due to temperature and humidity, but they did not take into consideration the aging issue. Papoola et al. [20] proposed a method to quantify and fit baseline variation caused by environmental variations. Their results showed good correlation between electrochemical sensors and reference data after extracting and correcting the baseline effect. Masey et al. [21] tested Aeroqual S500 systems including an electrochemical sensor for NO_2_. They tested three calibration protocols based on labeled data. They concluded that calibration model using labeled data from different periods is more performant than using labeled data collected solely from the first period. Wei et al. [17] proposed different linear models to compensate environmental factors and long term drift. However, data are collected during a short period (11 days) which is not enough time to study the long term drift. According to European protocol used to evaluate low cost sensors for air pollution monitoring, it is recommended to take some measurements each half a month for a period of 3 months to quantify long term drift [22]. Similar work was conducted in Sun et al. [23], where the long term drift was addressed using data collected over 28 days. Mijling et al. [24] calibrated an electrochemical sensor in field next to an air monitoring station during an 8-day period; their results showed that it is necessary to include temperature and relative humidity, by using a multilinear regression approach, to improve significantly the NO_2_ accuracy with R^2^ ranging from 0.6 to 0.9. They also highlighted that it is necessary to perform a full recalibration after a period of 2 months. Zhang et al. [25] proposed simple time series estimation methods to correct the baseline of metal oxide gas sensors. These methods are based on auto regressive moving average (ARMA) and Kalman filter. They showed that both models can realize the prediction of sensors’ long-term baseline in e-nose application, but the model based on ARMA has a more significant prediction of sensors’ long-term baseline effect.

In this work, we propose a linear and empirical method for long term drift correction based on data collected in field, during 6 months, which cover almost all the climatic conditions in this geographical region. The parameters of long term drift correction model are identified using particle swarm optimization algorithm (PSO). To the best of our knowledge, it is the first study that addresses an unsupervised drift correction method based on data gathered in field during 6 consecutive months.

The remainder of this paper is organized as follows. In the next section we present the experimental set up, the data collection and the pretreatment procedures. Section 3 deals with the input selection for the recalibration model and presents our empirical unsupervised recalibration strategy. First, we consider the effect of placing the sensors inside a monitoring station, in a controlled environment, on the estimation accuracy. Then we introduce our approach for drift correction. We show that it is possible to make unsupervised drift correction and to extend the calibration model for more than 3 months without any recourse to labeled data. Finally, we conclude this paper with a conclusion and perspectives.

## 2. Data Collection and Pretreatment

To measure the concentration of nitrogen dioxide in real conditions, we designed a device composed of an electrochemical sensor NO_2_-B41F provided by Alphasense LTD with its conditioning circuits, the gas exposure chamber and the data acquisition unit. 

### 2.1. Electrochemical Sensor Principle

The working principle of the NO_2_ electrochemical sensor is based on electrochemical reaction. When the gas passes through the filter, it creates a reaction in the electrochemical cell. The surface of the working electrode is the site for the first half reaction (oxidation) generating an electronic charge balanced by the second half reaction (reduction) that occurs at the counter electrode [19]. The reference electrode helps to maintain the potential of the working electrode at a defined value while the potential of the counter electrode varies in presence of the target gas, generating a balancing current proportional to the gas concentration to compensate the current generated at the working electrode [26]. A fourth electrode (the auxiliary electrode) is considered as a second working electrode, which has no contact with the target gas (Figure 1). It can generate a background current related the change of environmental conditions allowing to correct the working electrode current. New generation NO_2_ electrochemical sensors contain an ozone filter to forbid the access of this interfering gas to the electrochemical cell [27]. Alphasense guaranties the 80% of the original signal for 18 months of use. This lifetime is of course correlated to the environmental conditions such as humidity rate (dry air can reduce the electrolyte volume that influences the chemical reaction rate), gas concentration and interfering gas concentration that affect the electrode surface and filter capacity.

A potentiostat circuit for signal conditioning allows amplifying and converting the working and auxiliary electrode currents to voltage. Then, the concentration of the target gas is calculated as follows:(1)$$\left[NO_{2}\right]=\frac{\left(WE-WE_{0}\right)-\left(AE-AE_{0}\right)}{S}$$
where [*NO*_2_] is the concentration of the target gas; *WE*, *AE* are signals of the working and the auxiliary electrodes, respectively; *WE_0_* and *AE*_0_ are the total zero voltage offsets of *WE* and *AE*, respectively; *S* is the total sensitivity [mV/ppb], as calibrated in laboratory.

Equation (1) can be written:(2)$$\left[NO_{2}\right]=\frac{\left(WE-AE)+(AE_{0}-WE_{0}\right)}{S}$$

This is equivalent to:(3)$$\left[NO_{2}\right]=\left(WE-AE\right)*\alpha_{1}+\alpha_{2}$$
where *α*_1_, *α*_2_ can be determined by a linear regression replacing *1/S* and *(AE_0_ − WE_0_)/S* respectively.

Due to the difference between the surface of the working and auxiliary electrodes, it is recommended to assign an independent coefficient regression for each electrode signal. Therefore, it would be better to use Equation (4) rather than Equation (3):(4)$$\left[NO_{2}\right]=WE*a-AE*b+c$$
where *a*, *b* and *c* are regression coefficients obtained using multiple linear regression.

### 2.2. Sensor Data Collection

The sensor device is placed inside the measurement station managed by French air quality monitoring agency named ATMO Grand Est agency. This station is located beside the highway crossing the city of Metz, France. As shown in Figure 1, our device operates with a dynamic air-sampling mode using a pump and a mass flow controller (from Bronkhorst, France) placed on the sensor chamber exit to generate a constant and continuous airflow. This technique allows eliminating the influence of the wind speed variation. We set the airflow rate to 500 mL/min, in order to obtain the same airflow rate as the ATMO Grand Est NO_2_ analyzer (considered as reference instrument). This analyzer is based on chemiluminescence method according to standard NF EN 14211. The ambient air is led from the outdoor to both reference and our sensor device using inert pipes.

We collected data continuously over several months. Device operation started on July 2018. We note that inside the station, the temperature was controlled and fixed at 22 °C. The influence of the outside air temperature variation is reduced, but it is not eliminated. Therefore, we collected the temperature and relative humidity data to analyze their impact on the sensor response accuracy. Collected data present the voltages of the sensor responses with a sampling frequency of 200 Hz. Sensor responses are then averaged over a period of 10 s and recorded on a computer using Matlab software. Finally, collected data are averaged again each 15 min in order to comply with the reference data provided by ATMO Grand Est.

### 2.3. Data Series Pretreatment

Data collected during these 6 months of continuous operation, may contain outliers and irrelevant values due to technical issues affecting both our sensor device and the NO_2_ analyzer.

The first step of pretreatment aims to remove outliers in our dataset. To that extent, we fixed a minimum and a maximum threshold on data values. Data that do not belong to this interval are outliers to be eliminated. We use the Equation (1), where *WE*_0_, *AE*_0_ and *S* are provided by Alphasense. Then we fixed the minimum threshold at 0 and the maximum threshold at 2 times the hourly limit value (2 × 200 µg/m^3^) defined by European directives [22]. We should note that there is not a systematic method to fix the threshold, but our proposition can eliminate the extreme outliers, without removing any relevant data.

In this work, we investigate data that are collected over a period of 6 months. This is a very long period for continuous measurement where many issues and problems can arise. For example, the reference data can be affected by maintenance or re-calibration of the analyzer. In addition, some technical issues may cause missed data from our sensor device. Therefore, there is a need to identify these data, and to remove them from both sensor and reference datasets. After this treatment, a small desynchronization may appear between the two datasets. To solve this issue, we calculated the cross correlation between sensor data and reference data which reaches a maximum value when the two signals are maximally correlated. The location of the maximum value of the cross-correlations indicates time lag between two signals. Therefore, it is possible to synchronize the two datasets by removing delay [28]. We performed the data synchronization using the following algorithm: Compute the cross-correlations between the reference data *Xref* and the NO_2_ concentration *Xsens* obtained with sensor data series and the Equation (1):[*Cor*,*lag*] = *xcorr(X_sens_*, *X_ref_*); where *Cor* is the cross correlation vector and *lag* is the shift time vector of each shifted copy of *X_ref_*.The time lags are equivalent to the location of the maximum of the cross-correlations.[*M21*,*I21*] = *max(Cor)*; *M21* is the value of the maximum cross-correlation and *I21* is the rank (location) of this maximum.*Shiftedtime* = *lag* (*I21*);Synchronize the two data series *Xref* and *Xsens* by clipping the lags from both times data series.*Xref* = *Xref* (*Shiftedtime:end*);*Xsens* = *Xsens* (*1:(end-Shiftedtime*)).

Actually, as the missing data exist in many places of both time series datasets, it is not possible to synchronize the two time series by applying this algorithm on the overall dataset at one time. Therefore, we split the sensor and reference datasets in batches, and we applied the previous algorithm on each batch. Batches were then assembled to reconstitute the two data time series *Xsens* and *Xref*.

Figure 2 illustrates the complete sensor dataset (corresponding to NO_2_ estimated concentrations) and the reference data (corresponding to reference concentration) along with the time before and after synchronization. The zoom in a particular time can illustrate the shift between reference and sensor data.

## 3. Empirical Unsupervised Recalibration Strategy

Gas sensors show drift over time making the calibration model useless after a certain period. New calibrations are often needed. However, new calibrations or the application of methods such as calibration transfer, that we proposed in our previous work, need labeled data [29]. In this work, we propose a new approach aiming to correct long term drift without any recourse to labeled data. We analyzed first the impact of environmental factors on the sensor response and show the evolution of the prediction error over time. Then we proposed an empirical unsupervised drift correction method based on linear regression and particles swarm optimization.

### 3.1. Impact of Environmental Factors

This study considers only the impact of the temperature and the humidity of the air for two reasons. First, the wind speed and the air pressure do not affect our sensor response considering our constant air flowrate sampling. Secondly, electrochemical sensor as we used in this work, can be considered as greatly selective.

The impact of temperature and humidity on electrochemical sensors response was addressed recently in [30] using multiple linear regression.

To show whether temperature and humidity of the air influence the sensor response placed inside the station, we compared calibration models obtained by using the sensor response alone or with temperature and humidity parameters. Linear regression algorithm is applied on data collected during each month, from July to December to cover the most temperature and humidity conditions.

We use the root mean square error (RMSE) and the score of linear regression $R^{2}$ (coefficient of determination) as performance metrics to quantify the accuracy and the correlation between the estimated and the reference concentrations:(5)$$RMSE=\sqrt{N^{-1}\overset{N}{\underset{i=1}{\sum }}{\left({\tilde{y}}_{i}-y_{i}\right)}^{2}}$$
(6)$$R^{2}=\frac{\sum_{i=1}^{N}{\left({\tilde{y}}_{i}-\bar{y}\right)}^{2}}{\sum_{i=1}^{N}\;{\left(y_{i}-\bar{y}\right)}^{2}}$$
where *N* is the number of data; ${\tilde{y}}_{i}$, the estimated concentration; $y_{i}$, the concentration provided by reference instrument, and $\bar{y}$, the mean of reference concentrations.

The first calibration model uses only the NO_2_ sensor responses, without considering the environmental factors. For the other three models, we include these influential factors: (1) humidity, (2) temperature and (3) temperature and humidity variation, respectively. 

According to Table 1, during the summer season (July and August) there is no significant difference between these models in terms of accuracy. However, during the months of October, November and December, the model including temperature and humidity reduces the RMSE by about 1 µg/m^3^. This observation concludes that the influence of the temperature and humidity must be taken into account to address the climatic condition variations.

Figure 3 summarizes the results of Table 1 by illustrating the box plot of the RMSE and the score of linear regression R^2^ for each calibration model. We observe that the model, which considers temperature and humidity, is more stable comparing to other models and the gain in terms of accuracy is noticeable as shown in Figure 3 (the median of RMSE decreases from 8.3 to 7.9 and the R^2^ increases from 0.851 to 0.875).

Technically, using a long length of pipe tube to guide the air to sensor inside the station, in order to warm the air before exposing it to sensors, can reduce the influence of temperature variation. Beside this, a pump is necessary to suck air inside the station and to control the airflow. This eliminates the effect of wind speed. 

### 3.2. Unsupervised Empirical Drift Correction Algorithm

Our investigations led us to suggest the following experimental recalibration model for estimating the NO_2_ concentration.

First, we establish a calibration model that includes temperature and humidity values by applying multiple linear regression to calculate NO_2_ concentrations [*NO*_2_].

Then we propose a model to correct the sensitivity *coef1* and the baseline *coef2.* This model supposes that the degradations of the sensitivity and baseline are uniform over the time. Thus, the coefficients *coef*1 and *coef*2 are temporal functions. After testing different functions, we propose the following model that gives the optimum results.
(7)$${\left[NO_{2}\right]}_{corr}=\left[NO_{2}\right]*coef1+coef2$$
(8)$$coef1=\lambda_{1}\;\exp\left(\;\frac{time}{\lambda_{2}}\right)$$
(9)$$coef2=\lambda_{3}{*}^{\;}\frac{time}{\lambda_{2}}$$
where [*NO*_2_]*_corr_* is the corrected concentration, $\lambda_{1}$, $\lambda_{2}$ and $\lambda_{3}$ are correction parameters, and the *time* is the rank of data collected. These parameters are calculated using PSO algorithm, the reader can find a detailed presentation of the PSO algorithm in [31].

### 3.3. Experimental Validation and Implementation Guideline

To establish guideline for the practical implementation of this algorithm we considered several scenarios. First, we partitioned the 6-month data onto *n* batches of 2000 data (approximately 20 days) with *n* = 8. The aim of this partition is to create different scenarios and also to show the evolution of the RMSE from one batch to another. Figure 4 illustrates the evolution of the RMSE without applying any correction. Model calibrations built on the first batch keep providing a good concentration estimation for the first three batches then the RMSE value increases with time reaching 19 µg/m^3^ in the last batch. This is due to the changes in sensor baseline and sensitivity, and certainly to humidity and temperature variations.

We made different scenarios to evaluate our approach of correction model. In each scenario, we used the batch number *n_i_* to build the correction model which was tested on the succeeding batches (starting from *n_i +_*
_1_ to *n*_8_*)*. According to Figure 4, we started constructing scenarios from the batch *n*_4_ when sensor degradations were observed. The evaluation of each scenario is based on calculating the RMSE. Table 2 summarizes the different scenarios, and the procedure of our method is illustrated in Figure 5.

The results obtained using our correction model and applied in different scenarios are shown in Figure 6. The RMSE comparison when using different models (without correction or with our correction), shows the improvement of the concentration estimation quality in terms of precision. For the last batch (batch 8), regardless of the scenario order, the RMSE is reduced significantly from 19 to a value between 10.8 and 12.1 µg/m^3^. The correction model obtained by using batch 4 or batch 5 improves the concentration estimation of batches 6, 7 and 8 allowing to conclude that our approach can be used to extend the time between full recalibrations over more than 3 months.

Figure 7 shows the impact of our approach on improving the estimation of nitrogen dioxide concentration. The correlation between the reference concentration and the estimated concentration demonstrates that the corrected model can rectify both the baseline and the sensitivity. The slope and the intercept of linear regression between the reference and estimated concentrations were 0.81 and −8.4 before the correction, and 0.94 and −2.23, respectively, after the correction. Based on the scenarios, we suggest guidelines for long term drift correction method and we highlight its practical advantage regarding existing methods based on frequent full calibration.

To summarize, our proposed method supposes that linear gradual changes of the sensor sensitivity and baseline accrued over time. Therefore, we propose a time related correction model to consider the effect of baseline and sensitivity variation on the estimated concentration. First, we built a calibration model using data with known concentration obtained from a reference analyzer. Then we constructed a correction model using labeled data obtained from the reference analyzer. The parameters of the correction model are related to time. Once the model correction was established, we did not have recourse to label data, and we kept correcting the estimated concentration for more than 3 months (from batch 4 to batch 8). These results encourage the investment of low cost sensors which may partly replace the expensive air pollution monitoring instruments.

## 4. Conclusions

Electrochemical sensors are the most frequently used type of low cost gas sensors for air pollution monitoring, thanks to their small size, low cost, reproducibility, selectivity, linearity and low energy consumption. In this paper, we evaluated the performance of a four electrodes electrochemical sensor for in field nitrogen dioxide monitoring, and we proposed an empirical unsupervised recalibration method based on data collected over a period of 6 months, covering many climatic conditions.

First, we presented the data pretreatment, which is a very important step before establishing any calibration and/or correction model. Then we investigated the influence of the outdoor temperature and humidity on the response of NO_2_ electrochemical sensor placed in an air quality monitoring station. Thus, we compared different calibration models that consider or do not consider the temperature and relative humidity variations. The results show that there is a key difference in terms of accuracy even if the sensor is lodged in a controlled environment (temperature and air flow).

Then, we proposed an empirical unsupervised linear correction method by assuming the long term drift can affect uniformly the baseline drift and sensor sensitivity. We used PSO algorithm to identify the parameters of the correction model.

The performance of this method was assessed through experimental results based on various scenarios. The results allow to establish guidelines for the implementation of this method and highlight its practical interest.

The study presented in this paper reveals the effectiveness of the proposed drift correction approach. Nevertheless, low-cost gas sensor drift counteraction remains a challenging task, especially for air pollution monitoring where the pollutant gas concentrations and their variations are very low.

Our future work aims to further investigate the long term drift phenomenon and to elaborate formal criteria to help decision making in order to optimize the exploitation cost and the performance of outdoor air quality monitoring by using low-cost electrochemical sensors.

## Figures and Tables

**Figure 1 sensors-21-03581-f001:**
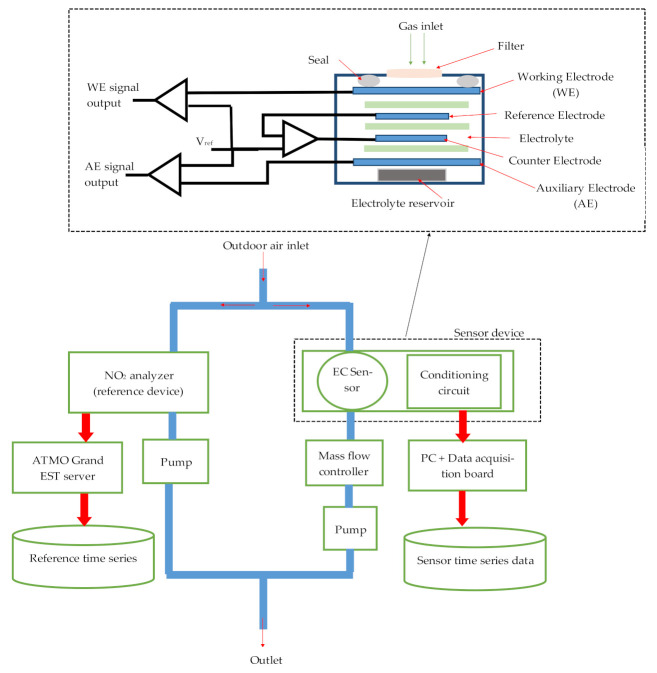
Experiment setup diagram and the schematic of four electrodes electrochemical sensor.

**Figure 2 sensors-21-03581-f002:**
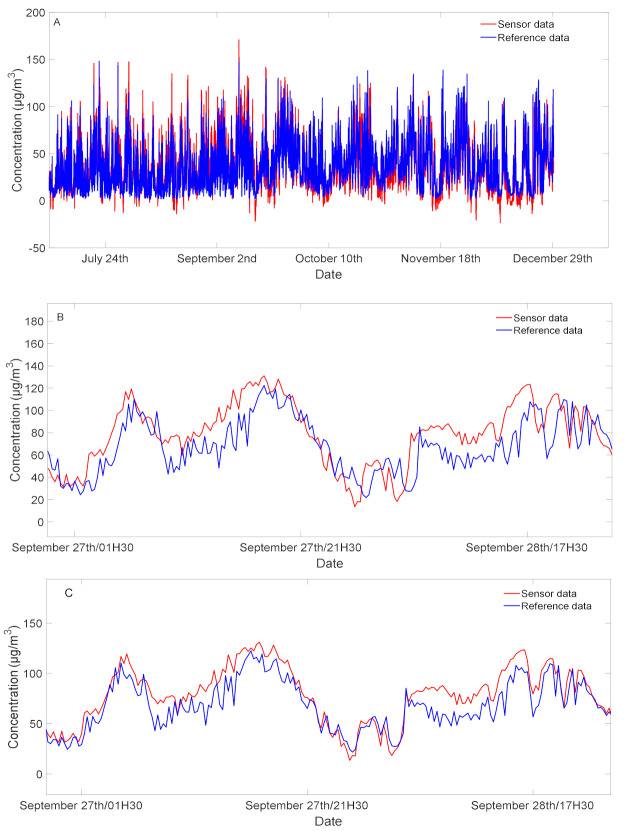
Synchronization between reference and sensor time series: (**A**) the entire dataset; (**B**) zoom to illustrate the time lags of the two times series; (**C**) zoom to show the synchronization after using cross correlation.

**Figure 3 sensors-21-03581-f003:**
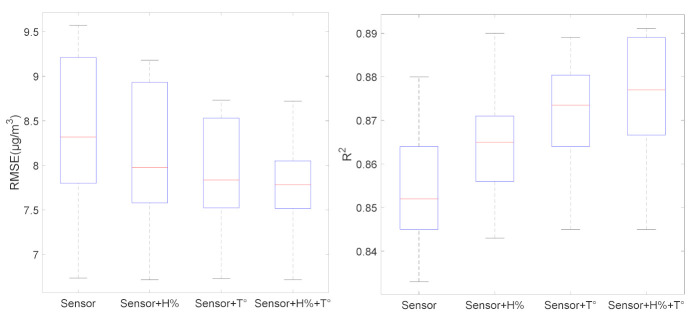
Box plot of the RMSE and R^2^ for each calibration model during a 6 month period (July to December).

**Figure 4 sensors-21-03581-f004:**
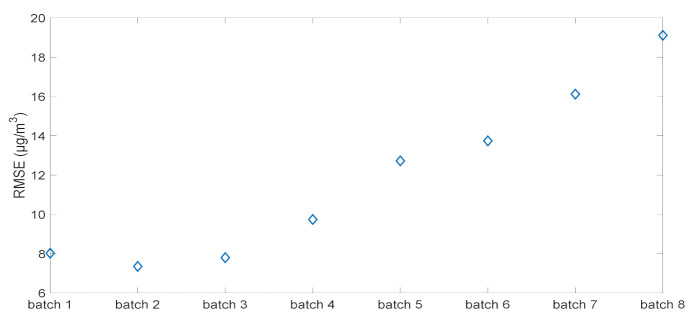
Evolution of the RMSE along with time in absence of drift correction.

**Figure 5 sensors-21-03581-f005:**
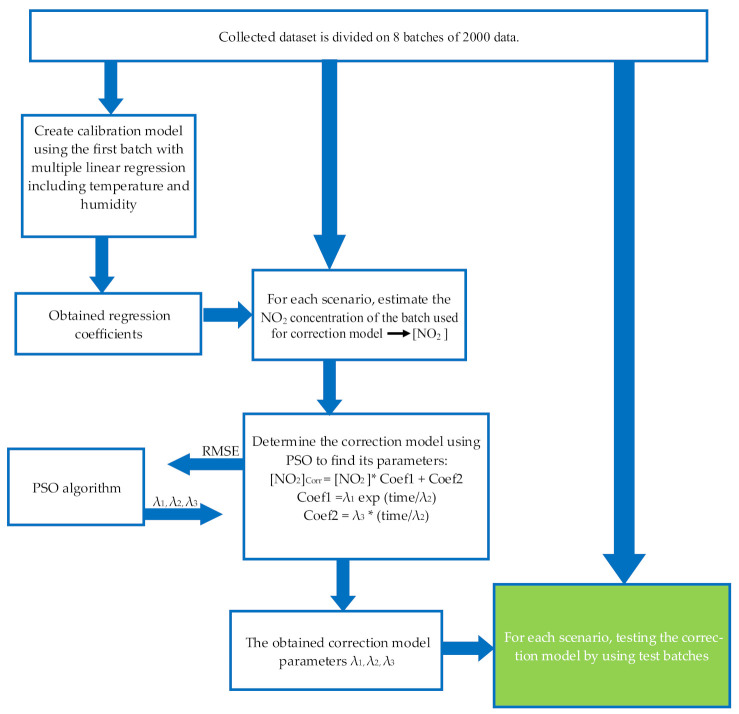
Empirical unsupervised drift correction algorithm procedure.

**Figure 6 sensors-21-03581-f006:**
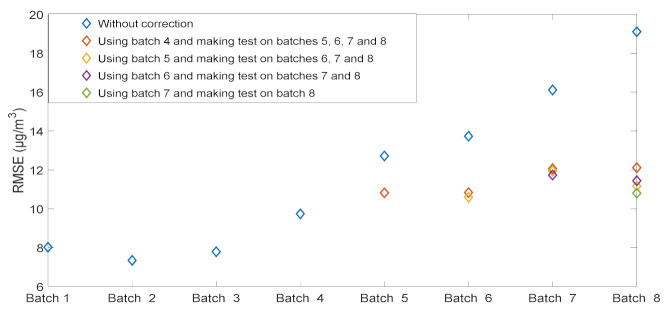
Comparison of the RMSE of nitrogen dioxide estimation before and after correction drift using different scenarios to evaluate our approach.

**Figure 7 sensors-21-03581-f007:**
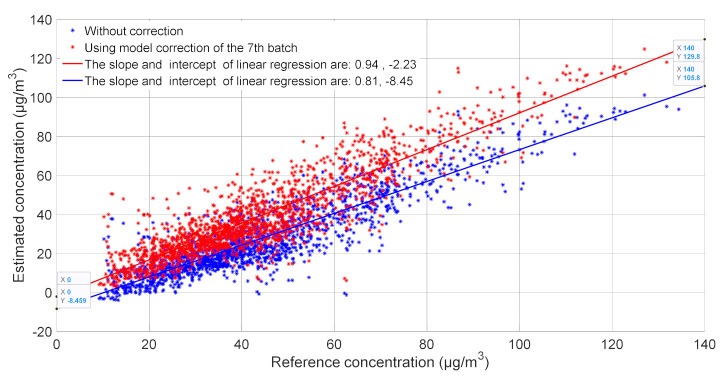
Correlation between reference concentration and estimated concentration of the 8th batch before (**blue**) and after drift correction (**red**) using the 7th batch.

**Table 1 sensors-21-03581-t001:** The accuracy of different calibration models obtained with or without environmental factors.

Model Using	Summer				Autumn						Winter	
	July		August		September		October		November		December	
	RMSE	R^2^	RMSE	R^2^	RMSE	R^2^	RMSE	R^2^	RMSE	R^2^	RMSE	R^2^
Sensor data	7.857	0.86	6.734	0.88	7.8	0.83	8.78	0.84	9.57	0.84	9.21	0.85
Sensor + humidity data	7.852	0.86	6.7154	0.89	7.579	0.84	8.1	0.87	8.93	0.86	9.18	0.85
Sensor + temperature data	7.851	0.86	6.73	0.88	7.523	0.84	7.82	0.88	8.73	0.87	8.53	0.87
Sensor + temperature + humidity data	7.807	0.86	6.688	0.89	7.517	0.84	7.76	0.88	8.72	0.87	8.05	0.88

**Table 2 sensors-21-03581-t002:** Scenarios used to validate the correction model.

Scenarios	Batch Used for Model Calibration	Batch Used for Model Correction	Batches Used for Test
1	1	4	5,6,7,8
2	1	5	6,7,8
3	1	6	7,8
4	1	7	8

## Data Availability

Not applicable.

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
