# Peer review of "Empiric Unsupervised Drifts Correction Method of Electrochemical Sensors for in Field Nitrogen Dioxide Monitoring"

_sensors, 2021, doi:10.3390/s21113581_

Round 1
Reviewer 1 Report
This work has been well planned, executed and analyzed. However, the manuscript needs to be revised before it can be considered for publication on the sensor.
- Please draw a complete diagram of the sensor installation, rather than simply replace it with a photograph, as shown in Figure 1.
- Whether to consider the cross interference of excessive gases, such as NO, CO2, etc.
- In the experimental test, how do you ensure the stability of the system test for the entire period of 6 months. The explanation on lines 143-175 in the article is not sufficient to explain the stability problem. There is no detailed data to support it.
- There are some spelling errors in the text, and there are repeated words, such as at the beginning of line 170, please review carefully.
Reviewer 2 Report
Some statements are not technically backed up by data. I felt the paper does not provide convincing arguments on the reasons for long-term drifts correction. What about EC sensor poisoning over time that will diminish the sensor sensitivity? Is this corrected by the hardware (reference sensor)?
Reviewer 3 Report
Please, find in the following my comments separated by Section.
Abstract
"Data are continuously collected onsite during a period of six months (July to December), covering almost all the climatic conditions." It could be added in which year, city and country data were acquired.
Introduction
"Several issues imped these systems" Is it a typo?
"However, data are collected during a short period (11 days) which is not enough to study the long term drift." How long should the experiment be then? How long is long enough?
The introduction talks about previous studies considering the effects of temperature and humidity on sensors. However, the Authors could provide more information on how exactly these parameters interfere with sensor electrical parameters (for instance, how does it change voltage, resistance or current) or its chemical compounds.
"These methods are based on the ARMA and Kalman filter. " Define ARMA.
"We show that it is possible make unsupervised drift correction for a period of three months without any recourse to labeled data. " According to what is writtten, one concludes that the study was performed for 6 months for data acquistion and for more 3 months for drift correction. If that is not correct, please, correct the text.
2. Data Collection and Pretreatment
"Therefore, it would be better to use the equation 4 rather than the equation (3)" Typo in (4).
"The device is placed inside the measurement station managed by French air quality service, ATMO Grand Est agency." Define ATMO.
"Thus, a pump and a mass flow controller (from Bronkhorst France) are placed on the exposure chamber exit, to generate a constant and continuous airflow by suction." Please, provide the specifications and suppliers of these equipment.
"we designed a 85 device composed of an electrochemical sensor NO2-B41F provided by Alphasense LTD". However, in Figure 1, it is written "e-nose". Is it just one sensor or a sensor array? When the Authors talk about the sensor, they should better describe how the e-nose was formed and provide more details on the sensors.
"Our device has been in operation since July 2018." It would be better to say "Device operation started in July 2018." In the same paragraph, the actions are described in the present, but the paper will be read in the future (at least, from 2021).
"Then we fixed the minimum threshold at 0 and 138 the maximum threshold at 2 times the hourly limit value (2*200 μg/m3) defined by European directives" Please, add a reference abput the directives.
"As the missing data exist in many places of both times series dataset, it is not possible to synchronize the two-time series by applying this algorithm on the hole dataset at one time." Many typos.
"Therefore, Therefore, we split the senor and analyzer datasets on many batches and apply the previous algorithm to each batch. " Many typos.
How exactly did the Authors split the data in batches? How were the batches defined and composed? 2000 data correspond to how many days of measurements?
3. Empirical Unsupervised Recalibration Strategy
"3.1. Impact of environmental factors:" No need for colon at the end.
"The first calibration model uses only the NO2 sensor responses, without considering the environmental factors. " Formatting error.
Tables and Figures starting with capital letters.
"This observation concludes that, the influence of the temperature and humidity must be taken into account when difference between inside and outside of the station is significant." When does it become exactly signicant?
"Technically, using a long enough tube to guide the air to sensor inside the station, in order to warm the air before exposing it to sensors, can reduce the influence of temperature variation. " How long is long enough?
3.2. Unsupervised Empirical Drift Correction Algorithm
"The reader can find a detailed presentation of the PSO algorithm in [29]." In any case, define PSO in the text.
3.3. Experimental validation and implementation guidelines
"We start constructing scenarios from the batch n4 allowing the adequate time for sensor degradation." How long is the adequate time?
"Model calibrations build on the first batch keep providing a good concentration estimation for the first three batches then the RMSE value increases with time and it reaches the 19μg/m3 in the last batch" Formatting error.
Description of Figures 5 to 7 are not clear. It is not clear how and why the latter batches provide better drift correction. There is no discussion comparing to the literature. Please, add references for the discussion and perform current state-of-the-art drift correciton techniques to compare to the proposed one.
Reviewer 4 Report
Review for
sensors-1188446
Empiric Unsupervised Drifts Correction Method of Electrochemical Sensors for in Field Nitrogen Dioxide Monitoring
I think, nonetheless, that the manuscript could be improved if the authors could address the comments and recommendations I listed below.
In the abstract, you may need to add some description about your methods and their novelty.
Line 90: I recommend you give a brief explanation about your sensor mechanism rather than just give a citation. Why it provides an electrical signal?
Figure 1: I recommend you give a brief explanation about the function of each component listed in your figure.
Figure 2: I do not understand your y-axis. Does your sample number relate to exposure time?
As you mentioned in your article, your collection period is about 6-month. Can you give a plot of NO2 vs. Time?
I'm wondering if your sensor will be affected by storms or heavy rainy days? Can you give some examples about it?
Line 220: Give a citation about the equation you applied here.
Some of your figures need error bars.
State more about your Long-term drift problems. What's the possible mechanism and how will you improve it?
Give a short description of the possible practical application. Is it possible to substitute the current high-cost sensors? Summary of your advantages.
Round 2
Reviewer 2 Report
low consumption - line 43 - power consumption?
"it recommended to take some" - line 68 - check English - " it is recommended ..."
Thus, eliminate the effect of wind speed. Line 240 - do you mean, this eliminates?
"To summary, our proposed method supposes that linear gradual changes of the .." line 292 - "In summary ..." "To summarize ..."??
"Thus, we compared different models, by including or not the temperature and/or relative humidity values." Line 318 - check English
A representative plot without the PSO would have been useful also.
What is the typical lifetime of an EC NO2 sensor?
The paper would have benefited some lab-based controlled environmental measurements data and analysis to validate the PSO algorithm.
Grids on Figure 7 would be useful.
It was not clear at what point the proposed algorithm would fail, where, either recalibration or sensor replacement will be required.
Reviewer 3 Report
The Authors have taken the opportunity to improve a lot the manuscript.
However, according to them, the novelty lies "To the best of our knowledge, it is the first study that addresses an unsupervised drift correction method based on data gathered in field during six consecutive months." It would be interesting to further describe the literature and, consequently, the innovation. For instance, how far advanced is this work from the rest? In addition, it would be interesting to apply other correcting methods so to compare and declare how better this one is.
In ay case, plase, perform the following minor corrections in the text:
1. Introduction
Page 2 line 48: error in "influences" (influence)
2.1 Electrochemical sensor principle
Page 3 line 110: error in "currant" (current)
2.2 Sensor data collection
Page 3 line 136: error in "referent" (reference)
2.3 Data series pretreatment
Page 5 line 191: error in "NO2" (subscript)
Page 6 line 196: please, properly punctuate the caption of Figure 2
3.2. Unsupervised Empirical Drift Correction Algorithm
Page 10 line 292: error in "to summary" (rewrite the beginning of the paragraph; use summarize).
As requested by the Authors, these are references that this reviewer considered as relevant for the submitted work. If the Authors consider them worth mentioning or subject to further discussion (in case the reference is already part of the text), please, go ahead and do it.
Sun, L.; Westerdahl, D.; Ning, Z. Development and Evaluation of A Novel and Cost-Effective Approach for Low-Cost NO2 Sensor Drift Correction. Sensors 2017, 17, 1916. https://doi.org/10.3390/s17081916
Yue Liang, Cheng Wu, Shutong Jiang, Yong Jie Li, Dui Wu, Mei Li, Peng Cheng, Wenda Yang, Chunlei Cheng, Lei Li, Tao Deng, Jia Yin Sun, Guowen He, Ben Liu, Teng Yao, Manman Wu, Zhen Zhou, Field comparison of electrochemical gas sensor data correction algorithms for ambient air measurements, Sensors and Actuators B: Chemical, Volume 327, 2021, 128897, ISSN 0925-4005, https://doi.org/10.1016/j.snb.2020.128897.
C. Chiu and Z. Zhang, "A novel gas sensor signal drift adjustment method based on controlled measurement," 2018 China Semiconductor Technology International Conference (CSTIC), 2018, pp. 1-5, doi:10.1109/CSTIC.2018.8369324.
Kovacs, Z.; SzöllÅ‘si, D.; Zaukuu, J.-L.Z.; Bodor, Z.; Vitális, F.; Aouadi, B.; Zsom-Muha, V.; Gillay, Z. Factors Influencing the Long-Term Stability of Electronic Tongue and Application of Improved Drift Correction Methods. Biosensors 2020, 10, 74. https://doi.org/10.3390/bios10070074
Lin, L., Zeng, X. Toward continuous amperometric gas sensing in ionic liquids: rationalization of signal drift nature and calibration methods. Anal Bioanal Chem 410, 4587–4596 (2018). https://doi.org/10.1007/s00216-018-1090-y
Wei, P.; Ning, Z.; Ye, S.; Sun, L.; Yang, F.; Wong, K.C.; Westerdahl, D.; Louie, P.K.K. Impact Analysis of Temperature and Humidity Conditions on Electrochemical Sensor Response in Ambient Air Quality Monitoring. Sensors 2018, 18, 59. https://doi.org/10.3390/s18020059
Cavallari, M.R.; Pastrana, L.M.; Sosa, C.D.F.; Marquina, A.M.R.; Izquierdo, J.E.E.; Fonseca, F.J.; Amorim, C.A.d.; Paterno, L.G.; Kymissis, I. Organic Thin-Film Transistors as Gas Sensors: A Review. Materials 2021, 14, 3. https://doi.org/10.3390/ma14010003
Reviewer 4 Report
Good to go.
Author Response
Thank very much for your valuable remarks and suggestions. they helped us a lot to improve this paper.